# Training Load Monitoring Considerations for Female Gaelic Team Sports: From Theory to Practice

**DOI:** 10.3390/sports9060084

**Published:** 2021-06-05

**Authors:** John D. Duggan, Jeremy A. Moody, Paul J. Byrne, Stephen-Mark Cooper, Lisa Ryan

**Affiliations:** 1Department of Sports, Exercise & Nutrition, Galway Mayo Institute of Technology, Galway Campus, Dublin Road, H91 T8NW Galway, Ireland; lisa.ryan@gmit.ie; 2School of Sport and Health Sciences (Sport), Cardiff Metropolitan University, Cyncoed Campus, Cardiff CF23 6XD, UK; jmoody@cardiffmet.ac.uk (J.A.M.); paul.byrne@itcarlow.ie (P.J.B.); SMCooper@cardiffmet.ac.uk (S.-M.C.); 3Department of Science and Health, Institute of Technology Carlow, R93 V960 Carlow, Ireland

**Keywords:** athlete monitoring, training load, Gaelic team sports, female athletes, internal load, external load, meaningful change

## Abstract

Athlete monitoring enables sports science practitioners to collect information to determine how athletes respond to training loads (TL) and the demands of competition. To date, recommendations for females are often adapted from their male counterparts. There is currently limited information available on TL monitoring in female Gaelic team sports in Ireland. The collection and analysis of female athlete monitoring data can provide valuable information to support the development of female team sports. Athletic monitoring can also support practitioners to help minimize risk of excessive TL and optimize potential athletic performance. The aims of this narrative review are to provide: (i) an overview of TL athlete monitoring in female team sports, (ii) a discussion of the potential metrics and tools used to monitor external TL and internal TL, (iii) the advantages and disadvantages of TL modalities for use in Gaelic team sports, and (iv) practical considerations on how to monitor TL to aid in the determination of meaningful change with female Gaelic team sports athletes.

## 1. Introduction

Athlete monitoring enables sports science practitioners to collect information to determine whether athletes are responding appropriately to training loads (TL) and the demands of competition [1]. Athlete monitoring can also support practitioners to minimize risk of excessive TL and optimize potential athletic performance [2]. Practitioners often collect a plethora of athlete monitoring data in relation to external and internal TL, fitness and fatigue that can illuminate whether athletes are adapting to their training programs, or whether they are at risk of overtraining, and an elevated potential for injury [3]. It is essential, therefore, that these data are collected and interpreted correctly, to inform decision making in relation to future planning and manipulation of the TL [4].

Over the past decade, there has been an exponential rise in both participation and professionalization of female team sports [5]. Notwithstanding this rise, there appears to be a disparity in research focused on monitoring TL in female and male team sports [6]. A consequence of this disparity is that practitioners have had to apply evidence developed on male athletes to female athletes, which has the potential to be erroneous [7]. Indeed, the collection and analysis of female athlete-specific monitoring data can provide practitioners with valuable information to support the development of female athletes and may lead to the professionalization of female team sports.

Ladies Gaelic football and camogie [kuh·mow·gee] are the most popular female sports in Ireland, and, in female Gaelic team sports, the premise of the two games are identical [8,9]. A team is comprised of 15 players, with the option of using five substitutes [10]. Each team includes a goalkeeper, two lines of three defensive players (full back and half back), two midfielders, and two lines of attacking players (half forward and full forward). Games at intercounty level comprise two 30 m min halves and games are played on a rectangular pitch 145 m in length and 90 m in width [11]. The primary objective of both these female Gaelic team sports is to outwit the defense and goalkeeper by sending the football (Gaelic football) or the solid leather slíotar [slit·er] (camogie) through the opposition’s goalposts (similar to rugby goal-posts), either below the cross bar for three points (goal) or above for a point [12]. In Gaelic football, the ball (diameter 680 mm, mass 480 g) is round, is like that used in soccer. The skills of the game include high catching, handling the ball, kicking the ball long distances, solo-running with the ball, blocking and intercepting [10]. In camogie, games are played with an ash stick, called a hurley, which is used to propel the slíotar (diameter 69–72 mm, mass 110–120 g) [12]. There are two major competitions at elite, intercounty level in female Gaelic team sports during the year, the National League and the All-Ireland Championship. The National League runs from January to April, whilst the All-Ireland Championship runs from May to September each year. During the competitive phase of the season, players may compete in games on weekly/fortnightly basis depending on progress through each competition.

Despite the amateur ethos of the game, elite intercounty female Gaelic team sport players complete up to five pitch-based, and resistance training-based, sessions each week [12,13]. Elite female Gaelic sports teams also have performance support teams to regulate and monitor athletes’ physical performance. Due to the fact that most of the research conducted to date has been in male Gaelic sports [14,15], the aim of this paper is to provide practitioners with practical considerations focused on implementing both internal TL and external TL in female Gaelic team sports environments. This will be accomplished by providing: (i) an overview of TL athlete monitoring in female team sports, (ii) a discussion of the potential metrics and tools used to monitor external TL and internal TL, (iii) the advantages and disadvantages of TL modalities, and (iv) practical considerations on how to monitor TL and determine meaningful change with female Gaelic seam sports athletes.

## 2. Materials and Methods

The following search strategies were used to located relevant article online through following databases which included EBSCO host, Web for Science, PubMed, Pub Med, SPORTDiscus and Google Scholar. Pertinent key terms in the search included training load monitoring OR monitoring OR training load AND female AND team sports AND internal load AND external load. Additionally, articles cited in the reference lists of acknowledged journals were manually searched and examined.

## 3. Training Load

Training load (TL) is often described as being either external TL and/or internal TL (Figure 1) [16,17], and has been described as an input training variable which can be manipulated to elicit a favorable training response [18]. Furthermore, TL is described as a stimulus experienced and responded to by an athlete before, during or after participation in the training process [16]. The quantification and monitoring of TL should form the basis of any athlete monitoring system [19]. The quantification of athlete monitoring data can aid in the interpretation and application of (1) iterative individualized athlete loading in preparation for competition, (2) specific load prescription, and (3) the subsequent physiological response to the load [18].

To improve athletic performance, athletes partake in systematic training which produces acute and chronic physiological adaptations [2]. A prescribed training dose will elicit specific physiological responses—this is known as a dose–response relationship [1,20]. These adaptations are associated with changes in performance (positive or negative) [21]. Whilst the response can be measured easily through positive change in performance metrics, or physiological adaptations, the dose is more difficult to measure [2]. It is important for practitioners to understand how a prescribed training dose will impact the specific physiological response in an athlete [22]. This dose–response relationship has been considered the holy grail for practitioners in team sports environments [23]. Due to the complex nature of team sports (physical, psychological, and tactical), the application of external TL and internal TL is crucial to understanding the athlete’s response to TL and can enable practitioners to modify training and recovery strategies (Figure 1) [1,20,22]. Additional challenges also arise for practitioners due to the large number of athletes in team-based sports, thus making it important to have a simple yet effective athlete monitoring system that involves both external TL and internal TL [24,25]. The relationship between internal TL and external TL has been conceptualized by Impellizzeri et al. [17] and is known as the training process. Figure 1 demonstrates how prescribed training and the unique physiological characteristics of the athlete combine to determine the internal TL which results in the training outcome.

TL monitoring techniques have become ubiquitous in male Gaelic team sports in the past decade [26,27,28,29,30]. The increased interest in TL has been due to the need to improve and individualize the training stimulus, to augment improvements in athletic performance and reduce the risk of overtraining and as well as the potential for injury [31]. This has been accelerated with the rapid evolution of technology in the sports performance domain [32]. A primary objective of sports training is to provide a stimulus that will potentially enhance sporting performance [19]. The quantification of training and competition load will enable practitioners to program activities to prepare and enable athlete recovery from the innate demands of game play [33]. TL is often manipulated at various times during the season to elicit a specific training response and enable the athlete to super compensate from a previous training stimulus [34]. The key skill of the practitioner, however, is to maximize the positive effects (fitness, readiness, and performance) and mitigate the negative effects (excessive fatigue, illness, and injury) of training [35]. Whilst, in theory, this sounds like a simple process, in practice, this is often a complex interaction of a myriad of physiological and psychological stressors [18,36]. TL data collection can also be useful to aid coaches in team selection and determining which athletes are prepared for the demands of competition [1].

### 3.1. Internal Training Load

Internal TL often referred to as the psychophysiological stimulus stressor that is imposed on an athlete due to the prescription of the external physical stimulus (Figure 1) [1,18]. For example, internal TL involves the quantification of the athlete’s response to the external load [18]. The measurement of internal TL can be subjective or objective [1,18]. These subjective, self-reported measures have been quantified previously in team sports through non-invasive methods include session rating of perceived exertion (s-RPE) and subjective questionnaire inventories [22]. Objective measures are not self-reported and include heart rate (HR) and blood lactate [22]. The ability to quantify internal TL may assist practitioners in providing greater understanding of the psychophysiological responses that occur through training and competition [16]. The adaptation which occurs through the training stimulus is a result of the internal TL which is determined by the external TL the athlete is subjected to [31]. The internal training response will dictate the training outcome; therefore, it is important that both external and internal TL measures are combined to provide a complete overview of the training process and response (Figure 1) [15].

### 3.2. External Training Load

External TL is described as the physical workload that is completed by athletes in training or match-play (duration, speed, distance covered) [22]. The prescription of external TL is determined by the individual athlete’s internal response (Figure 1). Consequently, the athlete’s characteristics such as their genetics, age, physical fitness capacities, and previous injury history will impact their external TL (Figure 1) [35,36]. One of the advantages of monitoring external TL is that it may enable practitioners to prescribe external TL metrics more precisely in future training sessions [31]. Some measurements which are commonly used in team sports include power output, global positioning system metrics, and accelerometer derived metrics [22]. There is a symbiotic relationship between internal TL and external TL, which is important in athlete monitoring as it can provide practitioners with the information to differentiate between a non-fatigued and a fatigued athlete [37].

## 4. Global Positioning Systems

Global positioning systems (GPS) technology was originally designed for military purposes [38]. It has now become common place in team sports settings to enable sports practitioners to collect comprehensive and real-time data during training or competition [39]. These devices can enable practitioners to monitor movement intensity and external TL, where monitored metrics include total distance covered, meters covered per minute, time spent at velocity thresholds, and the number of efforts at specific velocity thresholds [40]. Due to the increased professionalism of female team sports, there has been an increase in research analyzing the demands of the sports [41,42,43,44,45].

Despite the plethora of GPS research in the male versions of Gaelic team sports, research in female Gaelic team sports is still at the embryonic stage [26,27,28,29,30]. However, research using GPS analysis in camogie has demonstrated that in 60 min of match-play, elite intercounty players covered (mean ± SD): 5881 ± 906 m in total distance; 546 ± 259 m in high-speed running, and 183 ± 130 m sprinting [45]. Unfortunately, there appears to be no such peer-reviewed data for ladies Gaelic football. However, elite female Australian Rules Football (AFL) players, across the range of positions covered 6.0–7.0 km at 95.0–126.0 m·min^−1^ during 80 min of match-play [46]. Further research conducted in elite female soccer demonstrated that players cover between 9.5 and 10.3 km per game [47,48,49,50]. Practitioners should be cognizant of the fact that, whilst total distance covered may be a basic and ancillary indicator of the demands of the game, overemphasis of its worth may undermine the unique physical and physiological aspects of match-play [51].

To gain a more insightful overview of match demands, high-intensity activities need to be considered. In elite female soccer, Krustrup et al. [48] showed that these activities consisted of 5% of the total game time. Furthermore, there are distinct differences between competitive levels, where elite players cover between 1.5 and 1.7 km of high-intensity activities and this was between 15% and 30% higher than subelite players [49,50]. In camogie, full backs covered less total distance (m) and relative distance (m·min^−1^) compared to other positions [45]. Half backs and half forwards had higher sprint distances than all other playing positions [45]. Similarly, in elite international women’s soccer, central and wide midfielders covered superior distances at high speeds in comparison to central defenders, whilst full backs and forwards demonstrated similar high speeds [47,48]. Similar comparisons can be made with outcomes collected from other female team sports such as rugby sevens, AFL, and field hockey [41,44,52].

GPS data can also be used to objectively quantify TL on individual athlete’s, determine the demands of competition, examine individual positional workloads, establish training intensities, and monitor physiological fluctuations [53]. Modern GPS devices now include a variety of inertial sensors including magnetometers and gyroscopes, which, through a series of algorithms and filters, facilitate measurement of sport-specific metrics such as position, velocity, and acceleration, thereby enabling practitioners to quantify external TL [31,54]. Practitioners also need to be cognizant that measurement accuracy increases with increased frequency rates (10 Hz > 4 Hz > 1 Hz) [55]. Indeed, the accuracy of GPS seems to be greater during lower speed movements over longer distances with limited changes in direction [32]. This is an important consideration as higher-intensity movements are deemed to have significant implications for practitioners on performance, as they often occur during key phases in match-play [56]. Furthermore, these high-intensity movements are associated with higher injury risk, or when managed appropriately, may also have a protective effect on the athlete [57].

A further complication is related to the need to determine velocity and acceleration thresholds using absolute and individualized methods [51,58]. Absolute thresholds are easy to use and they enable practitioners to compare physical characteristics between players in different positional roles. Absolute thresholds have been used in male Gaelic team sports, where high-speed running velocity threshold has been defined as 17 km·h^−1^ and a very-high-speed running velocity threshold has been defined as 22 km·h^−1^ [26,59]. These absolute thresholds have been subsequently adopted and applied in the female equivalent of the game [45]. It is further suggested that employing male-related speed velocity zones thresholds to the female team sports context could result in an underestimation of external TL [60,61]. Female-specific high-speed running velocity thresholds have been recommended in soccer due to the physiological gender differences in physical fitness/capacity [42,50]. Absolute thresholds for high-speed running velocity and high-speed running velocity of between 16.0 and 19.0 km·h^−1^ and 20.2 and 22.5 km·h^−1^, respectively, have been recommended for elite female soccer players [50,60,62]. Indeed, research conducted with elite female soccer players advocated the use of >19.8 km·h^−1^ for high-speed running velocity and >25.1 km·h^−1^ for very-high-speed running velocity as generic thresholds [47].

The use of generic velocity thresholds, however, may not facilitate the monitoring of player-specific workloads because absolute thresholds might not demonstrate the true energetic demands of the athlete, potentially leading to the misinterpretation of external TL [63,64]. A possible solution to this dilemma is the application of individualized thresholds to quantify locomotive distances [64]. Individualized methods can be expressed in relation to both maximum aerobic speed and/or maximum sprint speed [65,66]. These thresholds can then be used by practitioners to determine individualized values which reflect both high- and very-high-intensity exercise modalities. Determining both maximal sprint speed and maximum aerobic speed can enable practitioners’ understanding of overall locomotor and energy profiles in both the aerobic and anaerobic energy pathways of individual athletes [66]. Previous research in AFL suggests estimating the athlete’s maximal velocity over distances of between 30 and 40 m [67]. If this maximal velocity changes in training or match-play, the value then becomes the individual athlete’s new maximal velocity [52]. However, there is little evidence to suggest that using individual thresholds is superior to generic thresholds when it comes to monitoring elite female soccer players [68].

### Accelerations and Decelerations

Accelerations and decelerations are important elements of match-play in team-based sports, contributing to increased levels of mechanical stress, and contributing to overall biomechanical load, which can significantly impact performance potential [36,69]. Both accelerations and decelerations occur frequently in team sports and these movements are energetically costly and taxing on the neuromuscular system [22]. Monitoring both metrics can aid the practitioner in evaluating external TL [70]. When using GPS, accelerations are measured when there is an increase in speed for 0.5 s that exceeds a maximum acceleration of least >0.5 m·s^−2^. By contrast, decelerations occur when there is a decrease in speed for 0.5 s that exceeds a maximum deceleration of at least >0.5 m·s^−2^, and are reported in specific zones [71,72]. Generic acceleration and deceleration zones are classified as low ±1–2 m·s^−2^, moderate ±2–3 m·s^−2^, and high < ±3 m·s^−2^, respectively [73,74]. Thresholds for accelerations and decelerations have been reported as between ±2.78 and 4.00 m·s^−2^ [75,76]. Metabolic power has been suggested as a method to determine the energy cost of accelerations and decelerations [77]. Metabolic power measurements are categorised from low (0–10 W·kg^−1)^ to maximum (>55 W·kg^−1^), equivalent distance (distance covered during steady-state running on a flat grass surface) and the equivalent distance index (ratio between equivalent distance and total distance) [78]. Whilst the concept of metabolic power has attractive practical applications, its validity has been scrutinised due to the possible error of measuring accelerations using positional systems [79,80].

## 5. Internal Training Load: Heart Rate

Heart rate (HR), the number of heartbeats in a set period of time, expressed as beats per minute (b·min^−1^), can serve as a marker of internal TL, enabling the practitioner to monitor internally in response to the external TL [81]. Measuring individual athlete HR can offer practitioners an opportunity to factor in the principle of individuality into TL monitoring [20,82]. HR will increase or decrease in response to the physiological demands of training and match-play, to maintain and deliver oxygen to the body, and can enable practitioners prescribe training intensities based on the known linear relationship between HR and oxygen consumption (VO2) across a range of submaximal workloads [83]. The application of HR measures during training and match-play can also ensure that athletes receive an adequate internal load stimulus, and they provide useful feedback in maintaining or improving fitness capacities in preparing for the demands of match-play [84].

It has been demonstrated that in elite female hockey players, the average peak heart rate (HR_peak_) was 198 ± 4 b·min^−1^, with a mean intensity of 95 ± 1% HR [52]. In female collegiate basketball players, competitive HR_peak_ frequently reached ≥85% [85], and in elite female football players, the average HR during competitive games ranged between 152 and 186 b·min^−1^, the equivalent ~80 and 90% of HR_peak_ [48,51]. In semi-elite soccer, females partaking in a 70 m min game (comparable with female Gaelic team sports) players spent most of match-play in HR zones between 60 and 75 b·min^−1^ and 75–85% HR_peak_ [86].

One suggested method to facilitate calculating TL is to use HR intensity across a session and multiply by its duration. This method has been defined as Training Impulse or TRIMP [87,88]. The purpose of TRIMP is to provide a quantitative measure of internal TL physiological intensity during training and match-play [89]. TRIMP may provide practitioners with a strategy in which the components of training can be quantified into a single arbitrary unit [90]. Several TRIMP HR measures have been suggested for use in team sports, including the summated heart rate zones (SHRZ), Bannister’s TRIMP, and Lucia’s TRIMP, each of which have distinct advantages and disadvantages regarding their use in the context of team sports [91]. However, the SHRZ model could be a pragmatic solution to embed HR monitoring into an applied setting [84,92]. The advantage of this model lies in the convenience in which data are collected which does not interfere with the athlete’s movements during training and match-play. Additionally, the duration of exercise is combined with HR intensity, predetermined using weighted heart rate zones, with higher intensity given a greater weighting [93]. In the SHRZ model, zone 1 = 50–60% HR_max_, zone 2 = 60–70% HR_max_, zone 3 = 70–80% HR_max_, zone 4 = 80–90% HR_max_, and zone 5 = 90–100% HR_max_ [92], and SHRZ is derived in arbitrary units (AU) as:SHRZ (AU)=(duration in zone 1×1)+(duration in zone 2×2)+(duration in zone 3×3)+(duration in zone 4×4)+(duration in zone 5×5)

In semi-professional male basketball players, the SHRZ demonstrated sensitivity in detecting increases in training loads across microcycles [93]. Despite the practical advantages of the SHRZ, further research is needed to validate the use of SHRZ in team-sports environments and whether the weighting factors lack intra-athlete sensitivity. Smaller SHRZ (~2.5% HR_max_) appears to be more sensitive in determining internal TL than the traditional SHRZ zones in team sport [94]. Bannister’s TRIMP incorporates resting HR (HR_rest_), maximum HR (HR_max_) and mean HR (HR_ex_), in an algorithm aligned with the acknowledged linear relationship between HR and blood lactate [BLa^−1^], estimates of which can be gathered during incremental exercise to derive an athlete’s internal TL [88]. Bannister’s TRIMP can be computed in arbitrary units (AU) as:$$\mathrm{TRIMP}\;\mathrm{training}\;\mathrm{load}\;\left(\mathrm{AU}\right)=\frac{(\mathrm{duration}\;\left(\min\right)\times \left({\mathrm{HR}}_{\mathrm{ex}}-{\mathrm{HR}}_{\mathrm{rest}}\right)}{\left({\mathrm{HR}}_{\max}-{\mathrm{HR}}_{\mathrm{rest}}\right)\times 0.64e^{1.67x}}$$
HR_ex_ = mean HR during exercise, HR_rest_ = HR at rest, HR_max_ = maximal HR, *e* = 2.712 and *x* = (HR_ex_ − HR_rest_)/(HR_max_ − HR_rest_) [88].

One of the limitations of Banister’s TRIMP is that a standardised lactate curve is used in response to exercise which does not account for the individual athlete’s response to the training mode and stimulus [91]. Other TRIMP models may be more appropriate for team sport athletes, and they may be more efficient in enabling practitioners to capture internal TL during training and match-play.

Derivation of Lucia’s TRIMP also acknowledges the inter-relationship between HR and [BLa^−1^] in estimating internal TL. This method requires data based on the individual’s response to an incremental exercise test [95]. The internal response is quantified using durations spent exercising in three weighted HR intensity zones based on fixed and individualized [BLa^−1^], of 2.5 and 4.0 m·Mol·L^−1^ [91]. The Lucia TRIMP is derived as:Training load=(duration in zone 1×1)+(duration in zone 2×2)+(duration in zone 3×3)
Zone 1 = HR linked to [Bla^−1^] <2.5 m·Mol·L^−1^, zone 2 = HR linked to [Bla^−1^] <2.5 and ≤4.0 m·Mol·L^−1^, and zone 3 = HR linked to [Bla^−1^] >4.0 m·Mol·L^−1^ [91].

Derivation of Lucia’s TRIMP factors in the individual athlete’s onset of blood lactate accumulation (OBLA = >4.0 m·Mol·L^−1^), and it allows for an individual quantification of internal TL through [Bla^−1^] and HR responses. However, this approach may lack practical application in a team sports environment because it requires an athlete to undertake a maximal, graded exercise test to determine [Bla^−1^] and HR variables [96]. Practitioners may also consider the application of Stagno’s TRIMPmod [97] and the iTRIMP [98] to female Gaelic team sport athletes. However, their use is limited, as they require individual blood lactate measures which may be impractical in an amateur, team-based setting.

The use of HR metrics can also enable practitioners to design training sessions to meet the demands of the game and provide aerobic stimulus [99,100]. Previous research has suggested that if athletes achieved between 7% and 8% of training time >90% HRpeak, this would provide sufficient stimulus for maintaining/increasing aerobic fitness [100]. From a female perspective, data have demonstrated that small-sided games in soccer elicited a higher HR response (>85 HRmax) when compared to results from medium- and large-conditioned games [99]. There are clearly some significant limitations in using HR for quantifying TL, including knowledge of technical proficiency and expertise in interpreting the results. HR is also a poor variable for measuring high-intensity activities such as resistance, speed, and power-based training modalities [20,81]. HR appears to be most appropriate for field-based training sessions in team sport athletes, and different TL modalities may be more appropriate for other components of training.

## 6. Training Session-Based Ratings of Perceived Exertion

The perception of effort is often used to monitor training in team-based athletes’, and it is also often used to evaluate exercise intensity [16,101]. The most common metric used to monitor perceptual load in team sports is session-based ratings of perceived exertion (s-RPE) which is a modified version of Borg’s ubiquitous 6–20 RPE scale [102]. The s-RPE requires athletes to subjectively rate the perceived intensity of a given training session using a scale of 0–10 [103]. The purpose being to gauge the athlete’s global rating of the intensity of the session [104]. The session load can then be derived as the product of the duration of the session/match-play and the session intensity using the individual measure of s-RPE in arbitrary units (AU) [105]:
Session load (AU) = duration of the training session (mins) × s-RPE (0–10)

Previous research has demonstrated that there are significant correlations (r = 0.50 to r = 0.85, *p* < 0.01) between s-RPE and HR metrics (Bannister’s TRIMP, Edwards’s TRIMP, and Lucia’s TRIMP) in professional male soccer players [16]. From a female perspective, the s-RPE TL has shown significant correlations (*p* < 0.001) with all training modalities in soccer players [105]. Furthermore, s-RPE TL coincided with HR-based measures (SHRZ and Bannister’s TRIMP) for the quantification of internal TL in female soccer players [105]. In female basketball players, s-RPE was shown to be a sensitive marker of internal TL during a period of intended over-reaching during a competitive training cycle [106]. In elite female soccer players, there was a high correlation (r = 0.78, *p* < 0.001) between s-RPE and TRIMP [107]. In female field hockey, during a congested competition period, elite players had an average daily s-RPE of 350 ± 58 AU [108]. In female futsal players, the mean s-RPE throughout the season was 320 ± 127 AU, reinforcing how s-RPE was sensitive to monitor in-season TL [109]. An important recommendation is to ensure reliability and validity of s-RPE, which can be achieved by standardising instructions and allowing for an anchoring procedure to familiarise the athlete with the s-RPE scale [110]. Due to the global intensity rating of s-RPE, practitioners are also advised to take the measurement at least 10 min post-training and up to 12 h following completion of the session [111,112,113]. The application of s-RPE is not without its limitations as it is only a gestalt measurement of internal TL and may not be sensitive enough to measure the entire array of performance induced psychophysiological constructs [66,114].

Additional metrics that can be considered such as training monotony and strain can provide practitioners with supplementary information on the individuals internal TL [104]. For example, the weekly TL (AU) is estimated by adding the s-RPEs from all the training sessions for the week. Previous research has demonstrated an average weekly TL of 2200 ± 300 AU in female soccer players [115]. This weekly load can also be used to estimate training monotony which is the weekly variation in session load, derived as the ratio of the daily mean TL and the standard deviation of the daily TL [102]. The standard deviation can be calculated based on the length of the specific microcycle. Furthermore, training monotony is thought to measure the amount of variation in the TL and, if the monotony is high, this could be an indication that there is little variation in the TL. It is advocated that a monotony score of >2 is associated with potential negative training outcomes [102,116]. Alternatively, training strain is the cumulation of training monotony and the weekly load. Previous research in team sports has established that during training periods of high strain and monotony, athletes are at potential higher risk of injury and illness [117,118,119]. These simple metrics may enable practitioners to determine the individual athlete’s risk of over-reaching and/or overtraining. In conclusion, the likelihood of overtraining is increased with weekly TLs over 4400 AU, monotony over 2.2 AU and strain over 6000 AU, respectively [102,116,120].

From a female athlete perspective, research has demonstrated that TRIMP is not influenced by the menstrual cycle during training. However, training monotony and training strain were higher in the follicular stage compared to the ovulatory phase of the cycle [121]. Indeed, the monitoring of menstrual cycle phases may provide additional feedback on the impact of training loads in female athletes. Advantages of using s-RPE to measure internal TL is that s-RPE requires minimal equipment to gather (pen and paper), it may be a valid method to gauge high-intensity sessions (e.g., resistance, sprint, and power) than HR methods and has been validated as an indicator of internal TL in female team sport athletes [122]. Another variation of RPE is differential RPE (dRPE). The dRPE enables the athlete to provide separate ratings of perceived exertion (e.g., breathlessness, s-RPE-B-TL × session duration) and peripheral exertion (e.g., leg muscle, s-RPE-L-TL × session duration) [123]. The dRPE may enable sport science practitioners to determine a more comprehensive quantification of internal TL in female team sport athletes [124,125]. This may be beneficial as respiratory and muscular exertion require differing quantities of recovery time and utilise alternate physiological adaptation pathways [36].

## 7. Athlete Self-Reported Measures

The use of athlete self-reported measures (ASRM) has become common practice in team sports as a method for measuring athletes’ levels of fatigue and readiness for training and competition [126]. The accumulation of stress and lack of recovery towards the end of the season in elite male soccer players has been reported to cause a decline in the recovery–stress balance [127]. These ASRM methods are often in the form of questionnaires and diaries which are used as simple and cost-effective ways to monitor the athlete’s response to training. However, as with all sport sciences measurements, their efficacy is reliant on how they are implemented and utilised [128]. ASRMs use measures of perceived well-being, including fatigue and psychological variables such as mood, which appear to be influenced by both training and non-training stressors [129]. It has been suggested that fluctuations in well-being are linked with potential over-reaching and/or overtraining [130,131]. Research suggests that female athletes report a different frequency of levels of psychosocial stress (recovery, and sleep quality) in comparison to male athletes [132]. Furthermore, female athletes appear to respond differently to similar TL than males and report higher perceived exertion, which may lead to poor recovery and self-efficacy [133]. Research that is centered around ASRM has come from valid and reliable measures such as the ubiquitous Profile of Mood States [134], Daily Analysis of Life Demands for Athletes [135] and the Recovery-Stress Questionnaire for Athletes [136]. Some of the limitations of these questionnaires include the lack of specificity to team-based sports, and the time burden on the athlete to complete the process. This has led to practitioners developing their own ASRMs to meet the demands of wellness monitoring to reduce the burden on the athlete and to increase compliance [126,137]. Consequently, reliability and validity of these ASRMs have come into question [138,139].

An example of a popularised ASRM is the Hooper Index [140,141], which comprises a five-point Likert questionnaire asking athletes to subjectively rate four items: mood (1 = very stressed to 5 = no stress), sleep quality (1 = poor to 5 = very good), energy levels (1 = extremely low to 5 = high/excellent), and upper and lower body muscle soreness (both separately scored, 1 = extreme soreness to 5 = no soreness). Scores for these are aggregated to give a readiness to train (RTT) score out of a maximum 100 which is purported to represent the overall stress in an athlete, with 100 being no fatigue/stress and an optimal level of readiness to train [142]. It has been further theorised that certain variables which the Hooper Index records may be more sensitive to determine internal TL. For example, a decrease in daily mood was found to be an important predictor of in-season injury in professional female football players [143]. From a Gaelic team sport perspective, it has been established that elite male Gaelic footballers experience a significant reduction in sleep throughout a training camp [27]. Furthermore, it was reported that RTT was a poor metric to measure in pre-training and competition preparation in elite Gaelic football players [144]. The researchers advocate that metrics such as sleep quality, sleep duration, and muscle soreness should be monitored when planning TLs [144]. Jefferies et al. [145] suggest that practitioners need to take a cautious approach when using ASMRs as they have been adapted from original psychometric questionnaires and currently lack validation. Therefore, ASMRs may provide a catalyst as a ‘conversation starter’ with the athlete, develop rapport and may improve autonomy between the athlete and their performance team [146]. Table 1 provides possible advantages and disadvantages of TL monitoring modalities.

## 8. Training Considerations

The training stress balance (TSB) involves monitoring the acute and chronic TL, where the acute TL lasts between 5 and 14 days and is suggested to represent freshness. The chronic TL is the rolling average which lasts somewhere between 4 and 6 weeks and represents the current level of fitness [87]. The TSB represents the interaction between the two variables indicative of fitness and fatigue [148]. If an athlete increases their acute CL too quickly and over the chronic TL, the risk of over-reaching and fatigue may be increased. [148]. Alternatively, if the acute TL is below the chronic TL, this may cause the opposite effect, where the athlete is fresher but possibly under-trained [149]. The aim, therefore, is to provide the athlete with an optimum amount of chronic TL with the sporadic increases in acute TL to provide intentional over-reaching or peaking [150]. TSB may provide practitioners with a good heuristic to measure how the TL is progressing and whether the TL is increasing too quickly. A caveat of any model is that training measures must be both reliable and valid, with the implication that if they are not, errors will result in subsequent TL data [151].

The acute–chronic workload ratio (ACWR) has become a popular method in male Gaelic team sports to monitor training load and injury risk [57,152,153]. The ACWR uses Bannister’s model to track changes in predefined acute (5–7 days) and chronic workload (21–28 days) using external TL and/or internal TL measures. It has been suggested that multiple units of TL can be used to determine an athletes ACWR, such as distance traveled, RPE, and high-speed running [24,154]. The ACWR is determined as the ratio of the acute workload and the chronic workload (training load accumulated during the current 7 days/training load accumulated over the past 21 or 28 days) [155]. It has been suggested that ACWRs in the ‘sweet spot’ between 0.8 and 1.3 are associated with a lower risk of injury. Conversely, injury risks are purported to increase with ratios below 0.8 and above 1.3 [22].

While the simplicity and practicality of deriving the ACWR seem an attractive proposition for practitioners, there are several limitations in using this model in team sports settings [156]. Practitioners need to collect enough longitudinal data which are specific to the cohort of athletes to which the ACWR is to be applied. Practitioners will also need to determine the most meaningful ratio windows (e.g., 7/27 days versus 6/21 days) that meet the needs of the sport [157]. The optimal ACWR need to be determined for each of the TL measurements associated with the ‘sweet spots’ which will differ from measure to measure. Importantly too, the use of ACWRs has recently been criticised for potentially producing false-positive outcomes. That is, ratios can be high, but players remain injury free [158,159,160]. Furthermore, the ACWR causes mathematical coupling, which results in potentially spurious outcomes [158].

### Interpretation the Data

The use of basic statistical tools may enable practitioners to make more informed decisions to enhance their athlete monitoring system [161]. For example, the TL data can be used to interpret the measures reliability, validity, and inferences for injury, illness, and athlete performance [4]. Reliability refers to consistency of the measurement used in testing or performance [162]. Additionally, the estimation of reliability can also enable practitioners to determine the degree to which the data provided are accurate. A common measure of reliability is a coefficient of variation (CV) derived as the ratio of the sample standard deviation (SD) and sample mean often presented as a percentage (i.e., CV% = SD/mean × 100). If the CV% is ≤10%, this is deemed acceptable in the scientific community. However, a more stringent cut off (≤5%) may be more appropriate in sports science testing contexts [162]. Validity, on the other hand, estimates whether the test/instrument measures what it intends to measure—it is an index of the specificity of measurement [34]. Importantly too, validity presupposes reliability, and therefore, the valid collection and interpretation of TL data are imperative to a successful athlete monitoring system and the ability to detect meaningful change [163]. The utilisation of TL data can aid practitioners in determining the outcome of the training program and provide information on the prescription of training. The individualization of the training process is also important as it provides practitioners with an insight into the various responses to the training stimulus and how that stimulus affects the individual adaptations [1].

Z-scores may provide the practitioner with a method of gauging within athlete changes and are determined by the number of standard deviation (SD) units a score is away from the mean for the sample (see Table 2) [34,164]. If an athlete’s z-score was +1.5, for instance, this would indicate that their score is 1.5 SD units above the mean for the sample [164]. This simple method may enable practitioners to make informed decisions relatively quickly and effectively [165]. Z-scores also enable practitioners to implement a traffic light system for athlete monitoring which can be easily interpreted by both playing and coaching staff [166]. Furthermore, they provide the opportunity to compare TL variables across both internal TL and external TL [167].

Practitioners often need to determine whether any change in TL data is meaningful (sensitivity). Smallest worthwhile change (SWC) has been suggested as a method to determine whether the change in performance is ‘real’ in relation to physiological or psychological outcomes [162,168]. The recommended SWC for monitoring TL in team-based sports is calculated as: 0.2 × between subject SD [168]. For suggested recommendations for SWC for both external TL and internal TL metrics, see Table 2. The typical error of measurement (TEM = standard deviation of difference scores ÷ square root of 2) can be used to determine true changes in performance when using a predetermined SWC [34,169]. The TEM provides practitioners with another method of expressing the error in a measurement [34]. It has been suggested that if the SWC is greater than the TEM, the practitioner can be confident that any changes are not due to error or noise associated with the monitoring tool [34]. Furthermore, practitioners can be confident that the change is real when the SWC is greater than the CV% [37].

Effect sizes can also be used to demonstrate the magnitude of change in TL variables. Effect size can allow practitioners to report data that can be easily understood regardless of the measurement used, which can be useful when presenting data to relevant stakeholders [170]. One of the most popular effect size indices is Cohen’s d which is derived as the ratio of the change in sample mean scores and the pooled SD between the two sample means [171]. This approach is the preferred method in sports science research because it is less influenced by sample size [172,173].
Cohen’s d = (mean 2 − mean 1)/pooled SD

Cohen’s d magnitudes of TL change can then be calculated (considered against the SWC) by using the following guidelines: trivial <0.2; small = 0.2–0.6; moderate = 0.6–1.2; large = 1.2–2.0; very large 2.0–4.0; extremely large >4.0 [174,175]. Indeed, this approach may enable practitioners in designing individual and optimize training programs across playing positions and levels in female team sports [171]. Table 2 provides an overview of how different statistical methods can be used as a part of an external and internal TL athlete monitoring system.

## 9. Conclusions

The ability to monitor TL in female team sports athletes is an important aspect of training program periodization. As athlete monitoring becomes more prevalent in female Gaelic team sports, this can ensure that athletes are prescribed the correct training dose at the right time, which is essential to increasing fitness and decreasing fatigue. When practitioners are selecting TL modalities, it is important to ensure that the methods used are both reliable and valid, and they are sensitive enough to potentially gauge and impact performance. A further consideration is the presentation of the data to the players and coaching staff, which should be easy to understand, non-invasive, and informative to the training process. The implementation of multifaceted athlete monitoring systems can provide practitioners with a number of benefits including the ability to (i) monitor the demands of training and competition; (ii) monitor athletes’ levels of fatigue; (iii) adjust and monitor individual athletes’ TLs in order to optimize performance; and (iv) reduce overtraining/injury risks. Figure 2 provides practitioners with a decision tree matrix to help to decide what athlete monitoring tool(s) are applicable to their environment. This review has presented several methods of monitoring both internal TL and external TL in female team sports with application specifically to female Gaelic team sports. This narrative review is not intended to be the panacea for athlete monitoring in female Gaelic team sports but may guide practitioners to adopt good practice when utilising TL monitoring modalities. It has also outlined the potential athlete monitoring tools which may be used as a foundation for practitioners to apply to female Gaelic team sports. Possible advantages and disadvantages of TL monitoring tools are also provided (see Table 1). Consequently, we believe athlete monitoring tools can bridge the gap between the art and science of athlete preparation, enabling practitioners to use the data to help inform the practice. From a female athlete perspective, the application of menstrual cycle monitoring is a further important area for future research and consideration. This will enable practitioners to determine whether menstrual cycle activity has an impact on performance and whether training needs to be modified during different phases of the menstrual cycle. Future research is clearly needed for more female-specific focused literature rather than utilising data from male team sports. Future research should focus on the impact of menstruation on individual training needs of females athletes, the incidence of non-contact injuries in female Gaelic team sport athletes and an enhanced understanding of female physiology to determine both positive and negative effects on performance. This research could inform recommendations and improve understanding on how to best support the health and performance of female Gaelic team sport athletes. Furthermore, female Gaelic teams sports are not as well financially supported as their male counterparts, and this is inevitably going to mean that the implementation of some of the athlete monitoring modalities outlined in the review will not be feasible for some teams. Figure 2 provides practitioners with simple, cost-effective methods to implement an athlete monitoring system which is underpinned by the principles of training and periodization.

## Figures and Tables

**Figure 1 sports-09-00084-f001:**
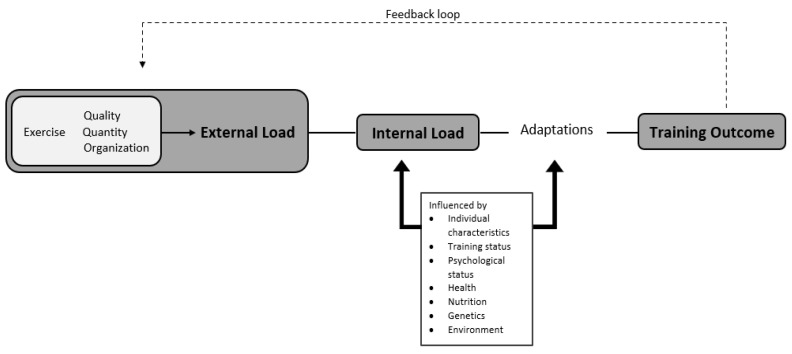
Training monitoring framework and quantifiable components for monitoring [18].

**Figure 2 sports-09-00084-f002:**
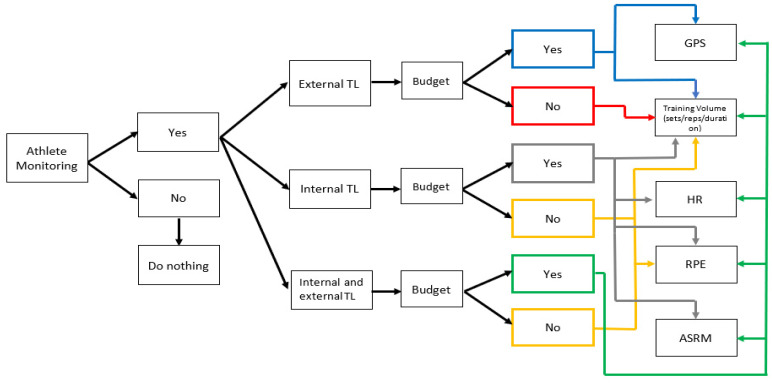
Decision matrix on the implementation of athlete monitoring in female Gaelic sports. TL = training load; GPS = global positioning system; HR = heart rate; RPE = rate of perceived exertion; ASRM = athlete self-reported measures.

**Table 1 sports-09-00084-t001:** Advantages and disadvantages of TL monitoring tools [147].

TL Monitoring Tools	Advantages	Disadvantages
**GPS**	External load metrics based on field performance Replicate physical game demands Live feedback Objectively determines the potential game demands	System cost Potential validity and reliability issues Time and expertise for data collection Influence of external factors such weather, HDOP signal, environment conditions
**HR Metrics**	Internal load metrics Easy to implement Instantaneous feedback	Cost Use in non-pitch-based sessions (gym, recovery, etc.) Collection of lactate profiles (Lucia, Stagno and TRIMP) Lack of consensus for best variable Potential validity and reliability issues around variables
**RPE**	Easy to implement Provides internal, subjective measure across all training modalities Validity and reliability for the metrics used Cost effective	Athlete questionnaire fatigue Athlete compliance Athlete can manipulate the data Single, gestalt measure of exercise intensity
**ASRM/RTT**	Easy to manage Potentially completed on a daily basis Non-fatiguing Cost effective	Athlete questionnaire fatigue Athlete compliance Athlete can manipulate the data Subjective information Lack of validity

GPS = global positioning system; HDOP = horizontal dilution of precision; HR = heart rate; RPE = rate of perceived exertion; ASRM = athlete self-reported measures; RTT = readiness to train.

**Table 2 sports-09-00084-t002:** Suggested TL monitoring strategies for female Gaelic team sports [34,82,147,176].

**External Measures**				
**Variables**	**Frequency**	**Objective**	**Analysis Method**	**Interpretations of Analysis data**
GPS TD Acc/Decel HSR VHSR MP	Field-based sessions	Measure of external field-based metrics		Avoid large spikes in week-to-week workload (10%) (Principle of progressive overload). Observe acute TL and chronic TL. **Daily Readiness:** SWC: TD: ~6% HSR Dist > 14.4 km/h: ~18% MP: ~14% **Between player normalization:** SWC: TD: ~2% Dist > 14.4 km/h: ~5% MP: ~5% Acc: 2%
Training Load	Weekly		*Z*-score relative to individual baseline measure	Avoid large spikes in week-to-week workload (10%) (principle of progressive overload). Observe acute TL and chronic TL. *Z*-score ≤ −1.5
**Internal Measures**				
**Variables**	**Frequency**	**Objective**	**Analysis Method**	**Interpretations of Analysis data**
HR	Field-based session	Measure internal field-based metrics	SHRZ, Bannister’s TRIMP	Avoid large spikes in week to week workload (10%) SWC ~−1%
Session RPE	Every session	Measure perceived exertion	*Z*-score relative to individual baseline measure	*Z*-score ≤ −1.5
Monotony	Weekly	Measure uniformity and training variation	*Z*-score relative to baseline score	*Z*-score ≤ −1.5
Strain	Weekly	Measure overall training load and monotony		*Z*-score ≤ −1.5
**Variables**	**Frequency**	**Objective**	**Analysis Method**	**Interpretations of Analysis data**
**Physio-Psycho measures**				
ASRM RTT (sleep quality, sleep duration, and muscle soreness)	2 to 3 per week	Measure overall wellness and quality of sleep, muscle soreness, fatigue, stress	Change in raw score per individual	*Z*-score ≤ −1.5 + 2.0 on measurement item = positive or negative change

GPS = global positioning system; TD = total distance; Acc = acceleration; decel = deceleration; HSR = high-speed running; VHSR = very-high-speed running; MP = metabolic power; SWC = smallest worthwhile change; TL: training load; HR = heart rate; SHRZ = summated heart rate zones; TRIMP = training impulse; RPE = rate of perceived exertion; RTT = readiness to train; ASRM = athlete self-reported measures.

## Data Availability

Not applicable.

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
