# Peer review of "Training Load Monitoring Considerations for Female Gaelic Team Sports: From Theory to Practice"

_sports, 2021, doi:10.3390/sports9060084_

Round 1

Reviewer 1 Report

My recommendations are the following:
To introduce in the abstract Methods and material and also to specify the most representative results and conclusions.
I recommend that you specify the sources of information, the databases.
How the data was collected in a separate section.

Author Response

To introduce in the abstract Methods and material and also to specify the most representative results and conclusions.

Thank-you, we have re-written the abstract, please see new Abstract, lines 12-22.

I recommend that you specify the sources of information, the databases.

Thank-you, please see new section 2, Methods and Materials, lines 70-75.

How the data was collected in a separate section.

Thank-you, please see new section 2, Methods and Materials, lines 70-75

Reviewer 2 Report

The authors provide an overview of the literature dealing with internal and external training loads athlete monitoring specifically in female Gaelic team sports, the potential metrics and tools used to monitor TL.

The general organization of the review is solid, references are comprehensive and updated, the experimental evidences are well interpreted.

I think that the article is of value and it is publishable in principle. However, I have some minor concerns that the authors can quickly solve.

The analysis of literature seems quite exhaustive.

Abstract: It is very confusing. It needs more depth. Please rewrite. Please clearly mention the suggested theory based on the current purpose.

Suggestions in the conclusions should be discussed regarding realistic effectivity.

Further limitations and useful suggestions for future research in this field need to be discussed.

Author Response

The analysis of literature seems quite exhaustive.

Abstract: It is very confusing. It needs more depth. Please rewrite. Please clearly mention the suggested theory based on the current purpose.

Thank-you, we have re-written the abstract, please see new Abstract, lines 12-22. We are also constrained by the 150-word limit of the Abstract

Suggestions in the conclusions should be discussed regarding realistic effectivity.

Thank you, please see new conclusion, section 9, lines 482-511.

Further limitations and useful suggestions for future research in this field need to be discussed.

Thank you, please see new conclusion, section 9, lines 482-511.

Reviewer 3 Report

I would like to thank for the opportunity to review the manuscript entitled "Training load monitoring considerations for female Gaelic team sports: From theory to practice". I find this paper well written and of great importance for two main reasons: the attention towards training load evaluation in female athletes, and the the attention towards "minor" sports that are often underrated and not sufficiently represented in sports science. I have no further comments and congratulate with the authors for this work.

Author Response

Thank you for the very kind words

Reviewer 4 Report

Thank you for submitting this work. I found the content very interesting and well developed. I believe several points are made in this manuscript that can translate to many, if not all, female sports. An area in need of greater address for the reasons you offer - safety and performance. 

Please find the comments or suggestions below as an attempt to assist in the refinement of the product. 

Abstract

Line 11 – Why are Team Sports capitalized? Perhaps ‘team sports’ is more appropriate. (seek consistency throughout the manuscript with this).

Line 11 – can you elaborate on intermittent, invasion-based a bit to allow a non-versed reader a chance to begin gaining a scope on the subject?

Line 15 – to help minimize…. We can reduce, but never eliminate

Line 17 – aims of this narrative are to

Line 18 – space between ii)discuss

Line 19 – TL to aid in the determination of meaningful….

Introduction

Line 26 – delete ‘their’

Line 27 – delete Furthermore…. Change to: Athlete monitoring can also….

Line 31 – and an elevated potential for injury

Line 37 – Change to read: A consequence of this disparity is…..

Line 40 – to support the development of female athletes and may lead to the professionalization of female team sports.

Line 59 – delete Because, change to Due to the fact that….

Line 62 – Delete ‘We will do this…” change to This will be accomplished by….

Line 64 – an extra space between the comma and the and before iv)

Also – there are 4 objectives here and only 3 in the abstract. Seek consistency with this.

Training Load

Line 72 – perhaps an addition of a statement as to how this quantification aids in interpretation and application? (just a suggestion)

Line 74 – This or A prescribed training load.

Line 79 – Delete Furthermore

Line 82 – change Further to Additional

Line 85 – an is known as the training process.

Line 104 – should this be indented?

Line 106 – change to as well as the potential for injury.

Line 108 – the term ‘match’ seems a bit out of context.

Line 117 – repeated terms at the beginning

Lines 117 – 125 – perhaps a reorganization of this paragraph is in order. I am not confident that the words convey a clear and concise message.

Line 141 – is games the appropriate term? Sport(s) perhaps.

Section 3.1 – odd alignment and spacing

Line 197 – indent

Line 224 – indent

Line 231 – indent

Line 247 – indent

Line 289 – indent

Line 299 – indent

Line 319 – typo after the [102]

Line 339 – should there be a formula offered for dRPE?

Line 340 – a shift in line spacing

Line 340 – indent

Line 344 – change way to ways

Line 379 – indent

Line 382 – delete ‘they’ it is vague. Change to a risk if ….may be increased.

General comment – there seems to be a lot of ‘however’s

Line 392 – travelled to traveled

Should a formula be offered for ACWR

Line 398 – Use a different term than ‘They’

Line 414 – 417 – Super long sentence

Section 7.1 – this is a great section! Well done.

Line 460 – indent

Three outcomes listed in the conclusion, but four offered earlier in the manuscript.

Reference 19 – no date?

Reference 20, 33, 34, 81, 87, 134, 171 – bold date

Author Response

I believe several points are made in this manuscript that can translate to many, if not all, female sports. An area in need of greater address for the reasons you offer - safety and performance.

Thank you for your kind comments.

Line 11 – Why are Team Sports capitalized? Perhaps ‘team sports’ is more appropriate. (seek consistency throughout the manuscript with this).

Thank-you, please see changes throughout the document

Line 11 – can you elaborate on intermittent, invasion-based a bit to allow a non-versed reader a chance to begin gaining a scope on the subject?

Line 15 – to help minimize…. We can reduce, but never eliminate

Thank-you, please see abstract, line 17.

Line 17 – aims of this narrative are to

Thank you, aims with an ‘s’ would suggest the use of is?

Line 18 – space between ii)discuss

Thank-you, please see abstract, line 19.

Line 19 – TL to aid in the determination of meaningful….

Thank-you, please see abstract, line 22.

Introduction

Line 26 – delete ‘their’

Thank-you, please see introduction, line 29.

Line 27 – delete Furthermore…. Change to: Athlete monitoring can also….

Thank-you, please see introduction, line 29.

Line 31 – and an elevated potential for injury

Thank-you, please see introduction, line 34.

Line 37 – Change to read: A consequence of this disparity is…..

Thank-you, please see introduction, line 40.

Line 40 – to support the development of female athletes and may lead to the professionalization of female team sports.

Thank-you, please see introduction, lines 42-43.

Line 59 – delete Because, change to Due to the fact that….

Thank-you, please see introduction, line 62.

Line 62 – Delete ‘We will do this…” change to This will be accomplished by….

Thank-you, please see introduction, line 64.

Line 64 – an extra space between the comma and the and before iv)

Thank-you, please see introduction, line 67.

Also – there are 4 objectives here and only 3 in the abstract. Seek consistency with this.

Thank-you, please see introduction, lines 67 -68.

Training Load

Line 72 – perhaps an addition of a statement as to how this quantification aids in interpretation and application? (just a suggestion) subsequent

Thank-you, please see lines 83-84.

Line 74 – This or A prescribed training load.

Thank-you, please see line 87.

Line 79 – Delete Furthermore

Thank-you, please see line 90.

Line 82 – change Further to Additional

Thank-you, please see line 95.

Line 85 – an is known as the training process.

Thank-you, please see line 98.

Line 104 – should this be indented?

Thank-you, please see line 118.

Line 106 – change to as well as the potential for injury.

Thank-you, please see line 120.

Line 108 – the term ‘match’ seems a bit out of context.

Thank-you, please see line 123.

Line 117 – repeated terms at the beginning

Thank-you, please see line 132.

Lines 117 – 125 – perhaps a reorganization of this paragraph is in order. I am not confident that the words convey a clear and concise message.

Thank-you, please see new internal load paragraph, 3.1, lines 132-42.

Line 141 – is games the appropriate term? Sport(s) perhaps.

Thank-you, please see line 158.

Section 3.1 – odd alignment and spacing

Thank-you, please see changes throughout the document.

Line 197 – indent

Thank-you, please see changes throughout the document.

Line 224 – indent

Thank-you, please see changes throughout the document.

Line 231 – indent

Thank-you, please see changes throughout the document.

Line 247 – indent

Thank-you, please see changes throughout the document.

Line 289 – indent

Thank-you, please see changes throughout the document.

Line 299 – indent

Thank-you, please see changes throughout the document.

Line 319 – typo after the [102]

Thank-you, typo has been deleted, Thank-you, please see line 158.

Line 339 – should there be a formula offered for dRPE?

Thank-you, please see formula for differential RPE, lines 353-534.

 Line 340 – a shift in line spacing

Thank-you, please see section 7, ASRM, lines 369-392. 

Line 340 – indent

Thank-you, please see changes throughout the document.

Line 344 – change way to ways

Thank-you, please see line 363.

Line 379 – indent

Thank-you, please see changes throughout the document.

Line 382 – delete ‘they’ it is vague. Change to a risk if ….may be increased.

Thank-you, please see line 405.

General comment – there seems to be a lot of ‘however’s therefore

Do you mean ‘therefore’? if so, this has been changed throughout the document. Thank-you, please see amended changes throughout the document.

Line 392 – travelled to travelled

Thank-you, please see line 414.

Should a formula be offered for ACWR

Thank you, please see lines 415-416.

Line 398 – Use a different term than ‘They’

Thank-you, please see line 421

Line 414 – 417 – Super long sentence

Thank you, this is written as two separate sentences

 Section 7.1 – this is a great section! Well done.

Thank-you

 Line 460 – indent

Thank-you, please see changes throughout the document.

Three outcomes listed in the conclusion, but four offered earlier in the manuscript.

Thank-you, please see lines 407-408.

Reference 19 – no date?

Thank-you, please see line 577.

Reference 20, 33, 34, 81, 87, 134, 171 – bold date

Thank-you, please see lines 581, 610, 611, 719, 732,736, 917, 925.

Identified journals were manually searched and examined.

Thank-you, please see new section 2, Methods and Materials, lines 70-75